# Parent-mediated play-based interventions to improve social communication and language skills of preschool autistic children: A systematic review and meta-analysis protocol

**Emre Deniz**, **Gill Francis**, **Carole Torgerson, Umar Toseeb** *

Department of Education, University of York, York, United Kingdom

* umar.toseeb@york.ac.uk

**Data Availability Statement:** All data relevant to this protocol is available in the provided link (https://osf.io/534f8/).

## Abstract

Early years interventions have shown to be effective in improving the social communication and language skills of autistic children. Therefore, various play-based interventions have been developed to support those developmental areas of autistic children. Although researchers have previously reported the overall effectiveness of different types of play-based interventions on the social communication and language skills of autistic children, no previous systematic reviews have yet evaluated the effectiveness of parent-mediated play-based interventions in preschool autistic children. The overarching aims of the study will be to (i) report the key characteristics and (ii) synthesise the results of studies evaluating parent-mediated play-based interventions targeting the social communication and language skills of preschool autistic children using experimental designs. A comprehensive search for and screening of the relevant studies published between 2000 and 2021 will be undertaken. To be included, studies will have to (i) use either a randomised control trial or quasi-experimental design, (ii) focus on preschool autistic children aged six years old or younger, (iii) deliver a play-based intervention in non-educational settings, and (iv) include at least one parent as the mediator of the intervention. Data extraction of all included studies will be undertaken using a specially devised template and they will also be assessed for risk of bias using an adapted form from the Cochrane Risk of Bias tool. The overall characteristics of the included studies will be reported and a narrative synthesis of the results of the included studies will be undertaken. A meta-analysis may be performed (if justified) to report the pooled effect size of the parent-mediated play-based interventions on the social communication and language skills of preschool autistic children.

**Trial registration:** The current study protocol was pre-registered with the international prospective register of systematic reviews (PROSPERO: CRD42022302220).

## Introduction

Although it is notoriously difficult to define the nature of play, as it is a wide-ranging term and intuitive understandings differ across disciplines, researchers generally agree on its following

**Funding:** The author(s) received no specific funding for this work.

**Competing interests:** The authors have declared that no competing interests exist.

characteristics: (i) not obligatory or serious, (ii) ungoverned by external rules, (iii) free and spontaneous, (iv) intrinsically motivated, (v) pleasurable [1–5]. Based on the aforementioned characteristics, the current study is guided by Rubin et al.'s [6] definition of play as activities that are intrinsically motivated, free to choose, spontaneous, purified from externally imposed rules, self-guided, non-literal, and require active engagement.

Play is an indispensable activity in children's lives as it contributes to many areas of child development including cognitive, social communication, language, emotional, and physical development. It has been suggested that symbolic play is effective in supporting perspective-taking [7], creativity [8], representational understanding [9], and theory of mind skills [10]. Play is instrumental in supporting children's various social communication skills such as joint attention and functional play [11], joint engagement [12, 13], social interaction [14], and social competence [15]. Regarding language skills, play has been found to improve the overall language [16] as well as expressive and receptive language skills [17, 18], the total words children produce, mean length of utterance, and linguistic complexity and diversity in at-risk preschool children [19]. Play is widely used as a therapeutic tool [20, 21] and has been found to have a significant role in decreasing children's internalising [22] and externalising problems [23, 24] and contributing to their healthy emotional development [25, 26]. Lastly, play provides children with limitless opportunities to be physically active in daily life, thus, helps them to develop muscles and gain physical strength [27].

Autism is a pervasive neurodevelopmental condition with a prevalence of approximately 1 in 44 children in the US [28]. Autism is characterised by persistent deficits in social communication, and language skills [29]. Autistic individuals may exhibit some or all of the following: deficits in social-emotional reciprocity, reduced sharing of interests, failure to initiate or respond to social interactions, deficits in nonverbal (e.g., eye contact, gestures, and facial expressions), and verbal communication. In addition, research have shown that autistic children have reduced levels of social communication [30–32], and language [33] abilities compared to their neurotypical peers. However, there is considerable heterogeneity as affected individuals have varied profiles of strengths and difficulties.

Many play-based interventions have been developed to support the social communication and language skills of autistic children [34, 35]. For instance, the Joint Attention Symbolic Play Engagement and Regulation (JASPER) intervention has repeatedly been found to be effective in improving joint attention, joint engagement, child-parent interaction, symbolic play, and language skills of autistic children [36–39]. Additionally, Pivotal Response Treatment (PRT) is effective in improving the social communication and language skills of autistic children [40–42]. Similarly, Child-Centred Play Therapy (CCPT) has also been found to be a significant intervention in increasing social participation and reciprocity skills and decreasing detrimental social behaviours [43–45]. Although different types of play-based interventions have been shown to be effective, the characteristics of such interventions vary mainly based on their primary focus, type of play, implementers, and mediators.

While the majority of play-based interventions have solely been built on the interaction between a child and interventionist, there has been a recent trend in the inclusion of mediators such as peers, teachers, and parents. This, perhaps, is not only due to the mediators' ability and availability in mediating such interventions but also targeting an improvement in both children's and mediators' outcomes within one intervention [46]. Unlike parent-mediated interventions, peer- and teacher-mediated interventions are often based on group play activities, conducted in educational settings, and are built on the elements of role-play and social reinforcement [47, 48]. Both peer and teacher-mediated interventions have been reported to be significant in improving social communication and language skills [49–51], as well as core autistic traits [52].

Parent-mediated interventions, however, differ from peer- and teacher-mediated interventions in the sense that they are often one-to-one activities between a parent and a child, take place in non-educational settings (e.g., home, clinics), and are commonly based on the principles of Mediated Learning Theory [53]. Parent-mediated interventions have been found to have significant positive effects on certain developmental skills of autistic children such as joint attention, social interaction, language, nonverbal communication, and adaptive functioning skills [54]. Additionally, a recent meta-analysis has reported a significant effect size for parent-mediated interventions on the communication and language skills of preschool autistic children [55]. Similarly, some researchers have found that parent-mediated interventions are effective in increasing language comprehension and child-parent interaction, as well as improving autistic traits in young autistic children [56].

Regarding the effectiveness of *parent-mediated play-based interventions*, JASPER has been found effective in increasing the joint engagement and play skills of preschool autistic children [46]. Additionally, the Play and Language for Autistic Youngsters (PLAY) intervention significantly improved social and emotional functioning and reduced autism severity in preschool autistic children [57]. In another study, a one-year Focused Playtime Intervention (FPI) has been reported to have significant positive effects on the expressive language skills of preschool autistic children [58]. A pilot study has reported that a parent-mediated Developmental, Individual-differences, Relationship-based model (DIR/Floortime) intervention significantly improved the functional and emotional skills and core autistic traits of preschool autistic children [59]. Although individual studies reported the effectiveness of parent-mediated play-based interventions, to the best of the authors' knowledge, no previous systematic review or meta-analysis has reported the characteristics and effectiveness of parent-mediated play-based interventions in preschool autistic children. Thus, the characteristics and overall effectiveness of such interventions remain unknown.

In the current protocol, the following elements of the proposed systematic review and meta-analysis are reported: (i) aims, (ii) research questions, (iii) study methodology (e.g., target sample, search strategy, inclusion criteria, predictor and outcome variables, data extraction, quality appraisal, and data analysis), and (iv) conflict of interest. The overarching aims of the study will be to (i) report the key characteristics and (ii) synthesise the results of studies evaluating parent-mediated play-based interventions targeting the social communication and language skills of preschool autistic children using experimental designs. To address these aims, the following research questions will be asked:

i. What are the key characteristics of parent-mediated play-based interventions?

ii. How effective are parent-mediated play-based interventions in improving the social communication and language skills of preschool autistic children?

iii. What factors, if any, mediate/moderate the impact of parent-mediated play-based interventions on children's social communication and language outcomes?

## Methods

### Study protocol and registration

The current study protocol was developed by following the Preferred Reporting Items for Systematic Review and Meta-Analysis Protocols (PRISMA-P) [60] checklist (S1 Table in S1 File) and pre-registered with the international prospective register of systematic reviews (PROSPERO) network to improve transparency, reduce potential bias and prevent duplication of another study (Ref: CRD42022302220). The results and supporting documents of the current

study will be made publicly available on the open science framework and submitted to a peer-reviewed journal for publication upon completion of the study.

## Study design

The current review will focus on the studies that applied randomised controlled trials or quasi-experimental designs with at least one control group or condition with the following characteristics: (i) a non-treated control group, with or without waitlist design, (ii) a control group that received treatment other than a play-based intervention.

## Target sample

The current review aims to search for and screen for studies that focus on preschool autistic children with social communication and language needs who were aged six years old or younger when the research was undertaken. Studies with a sample of children with a previously defined form of autism, such as Asperger's syndrome, autistic spectrum disorder, and pervasive developmental disorder-not otherwise specified (based on older versions of the DSM-V), will be included in this review.

## Intervention characteristics

The current review will include evaluations of parent-mediated play-based interventions that were conducted in non-educational (e.g., home, clinics, etc) settings and targeted social communication and language needs of preschool autistic children. Studies that did not report any information regarding the intervention setting will be treated as conducted in *a non-educational setting* and included in the sample as long as they were not delivered or mediated by teachers. Additionally, taking Rubin et al. 's [6] play approach into account, interventions that were not solely based on play, delivered during the playtime or within the playground will be excluded from this review. The latter criterion will be met in the following conditions:

i.  The intervention is a known and named play-based intervention (e.g., JASPER, PLAY, etc).

ii.  The intervention is a known and named therapeutic approach that uses play as a therapeutic tool (e.g., play therapy, child-centred play therapy, cognitive behavioural play therapy, Rogerian play, etc).

iii.  The intervention is a known and named developmental intervention that is built on the elements of play or consists of play-based activities such as joint attention, joint engagement, parent-child play (DIR Floortime, Pivotal Response Treatment, Early Start Denver Model, etc).

iv.  The intervention is not a previously validated and named intervention but is judged as a play-based intervention by the authors based on the characteristics of the intervention.

The last criterion will be met if "play is explicitly mentioned" [61] or if the intervention consists of parent-child play or child-led play activities. That is, any interventions that are not solely based on play are judged as *not play-based* (behavioural modification techniques, teaching and training activities, gaming, etc) will be excluded from the sample. Such interventions may include but are not limited to the following: Video modelling, video prompting, activity schedules, script fading, behavioural interventions, music therapy, virtual reality, gaming, and computer/video games, tablet applications.

## Predictor and outcome variables

The predictor variables will be parent-mediated play-based interventions that targeted social communication and language outcomes of preschool autistic children. The included outcome variables of the current study will be broadly set as all variables that are related to children's social communication (e.g., child-parent interactions, child-peer interactions, joint attention, joint engagement, joint interest, play skills, eye contact, social responsiveness, positive social interaction, initiation of social interaction, child's social functioning, etc), language skills (e.g., expressive and receptive language skills, vocabulary, the number of spoken words, the mean length of utterance, etc). Additionally, autistic traits will also be included in the outcome variables if one or more studies reported autistic traits (e.g., restricted repetitive behaviours, social affect, calibrated severity score, etc) as primary outcome variables.

## Study inclusion and exclusion criteria

The following criteria were pre-defined for the inclusion of the studies during the title and abstract screening phases [62]:

i.   Studies targeting children and youth younger than 19 years of age.

ii.  Studies with a sample of autistic children, including the ones who screened as "high likelihood for autism" on a validated autism screening measure, or children with a previously diagnosed form of autism (e.g., Asperger's syndrome, PDD-NOS, autistic spectrum disorder).

iii. Studies with randomised controlled trial (RCT) or quasi-experimental (QE) designs.

iv.  Studies that delivered a play-based intervention.

v.   Studies that are primarily focused on social communication, and language outcomes.

vi.  Studies published between 2000 and 2021, including grey literature.

vii. Studies that are published in English.

The predefined criteria were previously applied in Francis et al.'s [62] review which undertook an overarching in-depth review of children with social communication and language needs but reported only on mental health outcomes. The current review is undertaken by the same team and uses the same predefined criteria to study social communication and language outcomes using the same data identified from the database screening phase described in Francis et al. The inclusion criteria will be further narrowed at the full-text screening stage based on the focus of this review to select:

Studies targeting preschool autistic children aged six years old (72 months) or younger.

Studies including at least one parent as a mediator. Studies with multiple implementers will also be included if implementers are placed in separate intervention groups alongside a non-play control group (e.g., intervention group 1 = parent-mediated intervention, intervention group 2 = sibling-mediated intervention, and a control group).

Studies that were conducted in non-educational settings (i.e., home, nursery, clinic, etc).

Additionally, the following exclusion criteria will be applied at the full-text screening stage:

Studies will be excluded if the upper age limit of their sample was greater than six years old (72 months) unless the authors separately reported the outcomes of those aged six years old or younger.

Studies will be excluded if the participants in the sample were not formally diagnosed with or screened for autism.

Pre-experimental studies with no control groups will be excluded.

Studies that compared the effectiveness of two or more play-based interventions will be excluded unless they provide a control group that received a non-play-based intervention,

Studies that delivered a play-based intervention that does not meet the concept of play, that is defined in the current review, will be excluded.

Studies will be excluded if there are multiple implementers in a single intervention group (e.g., intervention group = parent and sibling mediated play intervention and a control group).

## Study selection procedure

As mentioned previously, the current study is part of an overarching review [62]. In the overarching review, 110 studies that reported the effectiveness of play-based interventions on the social communication and language skills or mental health of children with social communication and language needs were found after the full-text screening. The previous in-depth review reported on the (n = 10) studies that targeted mental health outcomes. The remaining studies (n = 100) are the starting point for identifying relevant studies for this in-depth review on the effectiveness of play-based parent-mediated interventions on *social communication* and *language* outcomes. Additional searching and screening will be carried out to capture recently published studies post the completion of Francis, et al. The current review, therefore, comprises three screening phases, outlined in Fig 1.

**Phase I: The overarching review.** In phase I, the authors carried out an overarching review, guided by Cochrane [63] and PRISMA-P [60]. Searching was undertaken on the following databases: Ebscohost (including ERIC), Ethos, ProQuest, PsycINFO, Pubmed, Scopus, Web of Science, and Google Scholar. The following predefined search strings were used to capture the studies that met the inclusion criteria: "Play", "speech", "language", "communication" "autism", "autistic", and "Asperger's" and named play interventions (e.g., JASPER, PRT, CCPT). In addition to the master terms, Boolean search strategies were also adopted by combining those pre-specified master keywords with some useful operators such as "AND", "OR", "NOT". and "quotation marks".

The database screening was conducted by one author (ED) using the search strings which resulted in 1785 potentially relevant studies after duplicates were dropped. Title and abstract screening was then carried out by two authors (ED & GF), independently. The two authors agreed on the inclusion of n = 366 studies, with a high inter-coder agreement (κ = .88 (95% CI, .85- .91). Disagreements were discussed and resolved within the research team. Lastly, a double screening of the full texts, by the same authors, indicated n = 100 studies addressing social communication and language outcomes and, within those 100 studies, n = 10 addressing mental health outcomes. The studies addressing social communication and language outcomes are the focus of this in-depth systematic review. In the current study, therefore, the pre-identified studies addressing social communication and language outcomes will be included and rescreened according to the relevant inclusion criteria. More information regarding the screening for Phase I is reported in the previous review [62].

**Phase II: Update.** Since Phase I comprises the studies that were published between 2000 and January 2021, an additional search for relevant studies published in 2021 will be carried out on the aforementioned databases and by using the same search strings (S2 Table in S1 File). The database search will be conducted by the first author (ED) alongside a

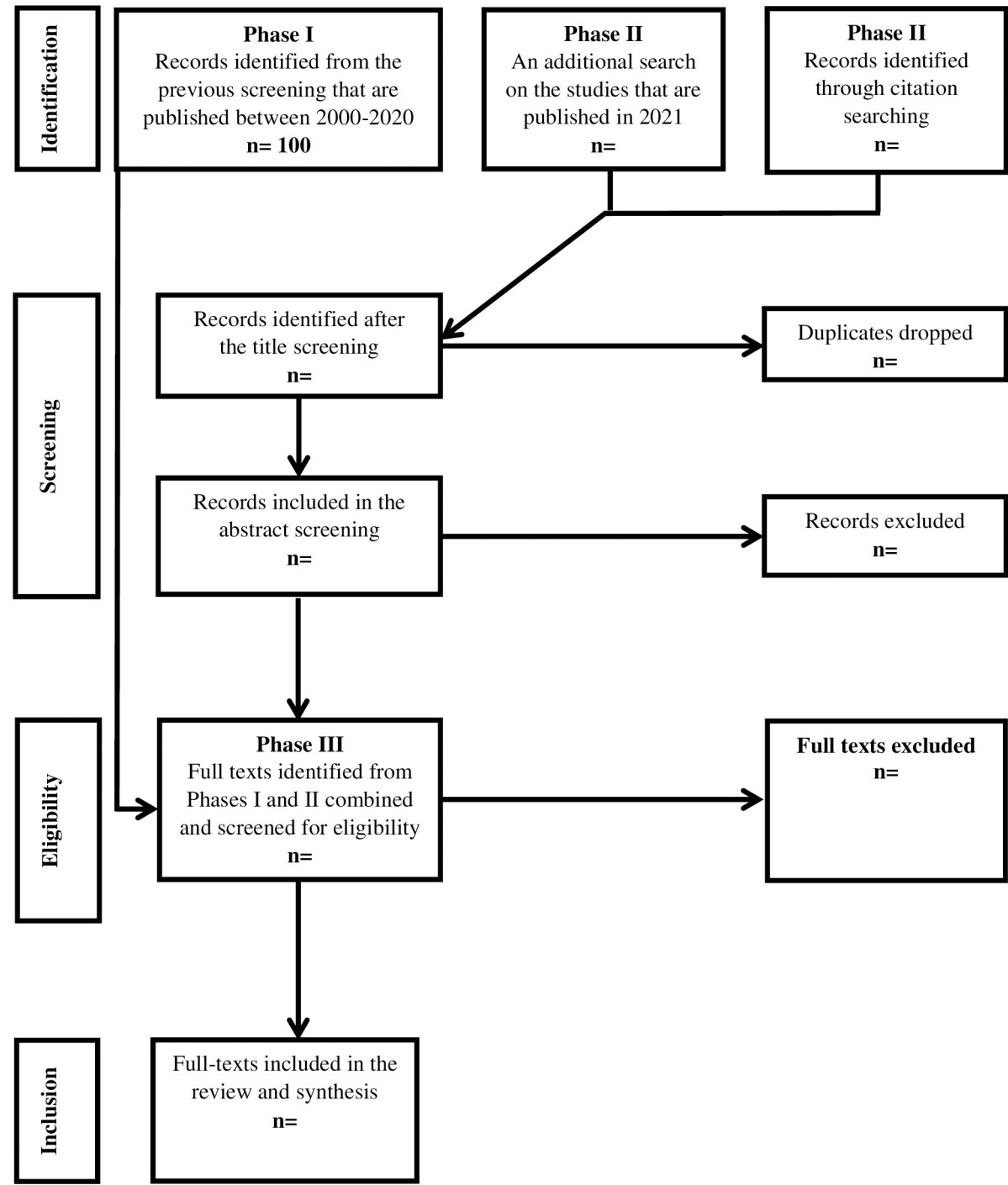

**Fig 1. Prisma flow diagram for records of literature screening.**

comprehensive citation search of the previous systematic review and meta-analysis on parent-mediated interventions in autism [39, 51, 54–56, 64–73]. The identified studies will be merged and downloaded into Mendeley Reference Manager software for performing the title, abstract and full-text screenings. The title and abstract screenings will then be conducted by two authors, (ED & GF), to make sure no eligible study is overlooked. Any discrepancies between

the two coders, at the title and abstract screening, will be resolved by the third author's opinion (CT). Lastly, the full-text screening will be carried out by two authors (ED & CT), independently, and disagreements will be resolved by the inclusion of a third reviewer (GF), prior to the final decision.

**Phase III: Current in-depth review.** In Phase III, firstly, the results from Phase I and Phase II will be merged, and duplicate studies will be dropped. Secondly, all studies will be rescreened by the four authors according to three additional eligibility criteria for this in-depth SR: (i) studies focusing on preschool autistic children aged six years old or younger, (ii) interventions that were conducted in non-educational settings, (iii) interventions that include at least one parent as a mediator. Eligible studies will then be included in the review and carried over to the data extraction stage.

## Data extraction

**Descriptive data.** Information related to intervention types, play approaches, sample sizes, experimental designs, etc, will be extracted to report the included study characteristics.

**Quantified outcomes.** Quantified results related to the social communication and language skills of autistic children will be extracted from the included studies for the narrative synthesis and meta-analysis (if justified). The authors will independently double data extract all quantified (primary and secondary) outcomes (not exploratory outcomes). All outcome measures will be reported. Should a meta-analysis be deemed appropriate, the authors will extract one outcome for each domain: social communication, language, and autistic traits. We will report one outcome for each domain using the following pre-specified algorithm: the pre-specified primary outcome(s) (where provided); the first outcome reported in text or table in the paper; if the first outcome is a subscale measure, the authors will use the total score of the sub-scale instead. Quantified outcomes will include numbers, means, standard deviations, standard errors, confidence intervals, and reported effect sizes that are reported regarding the effects of the delivered intervention on the social communication and language skills as well as autistic traits of preschool autistic children.

All data will be double extracted using a specially devised data extraction form (S3 Table in S1 File). The first author will extract the data from all included studies and each of the other authors (GF, CT, & UT) will extract data from a third of the sample studies. Any discrepancy between the coders regarding the extracted data will be discussed and resolved. If two coders are not able to come to an agreement, the third and fourth coders will be consulted. Upon completion of the data extraction, all numerical data will be extracted by using the quantified outcomes form (S4 Table in S1 File).

## Quality appraisal

The methodological quality of the included studies will be assessed by all four authors with the first author assessing all (ED) and each of the others (CT, GF, & UT) assessing a third of the sample, independently. The authors will be using two risk of bias (RoB) tools one of which is for randomised control trials and the other for quasi-experimental designs (S5 Table in S1 File). Both RoB forms were previously adapted [62] from the revised Cochrane Risk of Bias Tool for Randomized Trials (RoB 2) [74]. and Cochrane Risk of Bias Tool in Non-randomized Studies of Interventions' (ROBINS-1) [75]. Each tool has five domains that assess the following items: random allocation and concealment or allocation, blinding to the intervention, appropriate measurement, reported and selected outcomes, a prior analysis plan and a protocol.

Studies will be categorised under three categories: (i) low risk of bias, (ii) some concerns of bias, and (iii) high risk of bias by summarising the scores from five domains. The overall RoB

will be judged as *low risk of bias* if the study is judged to be at low risk of bias for all domains, *some concerns* if any domains are judged as some concerns, and *high* if any domains are judged as high risk of bias [74]. Although the level of risk of bias will not result in the exclusion of any studies from the review's sample, studies with a high risk of bias will be excluded from the meta-analysis (if undertaken). Furthermore, sensitivity analysis may be conducted to assess whether the methodological quality of studies (e.g., low risk of bias, some concerns) showed any impact on the reported effect size of the interventions. Additional subgroup sensitivity analyses may also be used to check if intervention effects vary by research design (RCT or QED) if there are sufficient relevant studies.

## Data analysis and synthesis

All statistical analyses will be conducted using statistical software R. If the interventions and outcomes are judged to be sufficiently homogeneous, a meta-analysis may be performed to report the overall effect sizes of the play-based interventions on the social communication and language skills of preschool autistic children with social communication and language needs. To do this, the reported outcome variables will be categorised under three umbrella terms: (i) social communication skills, (ii) language skills, and (iii) autistic traits. Also, an overall effect size may be reported by calculating the standardised mean difference (*Cohen's d*) on all reported social communication, language, and autism severity outcomes. Confidence intervals (CI, 95%) and standard errors (SE) will also be reported alongside the reported effect sizes. Additionally, statistical heterogeneity may be assessed using either *Cochran's Q ($x^2$)* or $I^2$ depending on the sample size of the meta-analysis. Given that the data on study outcomes, in this context, is likely to be heterogeneous, a random-effects analysis may be conducted to report the degree of variations between the study outcomes.

## Missing data

All included studies will be checked in terms of missing data. In case any missing data is found, the research team will discuss whether the missing part of the data affects the reported outcomes, if so, three solutions may be applied to deal with the missing data. First, the research team will try to contact the studies' corresponding authors to ask if the missing data could be shared. Second, the research team will try to impute the missing data by using other-reported quantified outcomes. For instance, in cases where the standard deviations (SD) are not provided, the research team will use the mean and sample size to calculate the SDs. Third, if corresponding authors do not respond and the reported information in the study is insufficient to impute the missing data, the study will be excluded from the meta-analysis, if undertaken.

## Discussion

Play has a significant role in contributing to children's development [76]. Therefore, various play-based interventions have been developed and used to support the social communication and language needs of autistic children [11–19]. Although many reviews and meta-analyses have reported the characteristics and overall effectiveness of play-based interventions on autistic children's social communication and language skills [39, 51, 54–56, 64–73], to the best of the authors' knowledge, the current study will be the first to evaluate the effectiveness of parent-mediated play-based interventions conducted in non-educational settings on the social communication and language skills of preschool autistic children. Hence, the review will shed new light on the role of parental mediation in the effectiveness of play-based interventions.

It is anticipated that the current study will have many strengths and some potential limitations. A major strength is that having a pre-registered study protocol that comprises an apriori

analysis plan will minimise any potential for bias in study selection and identification of outcomes. In addition, the design, conduct and reporting of the current study will be guided by the PRISMA statement and an adapted version of the widely used Cochrane quality assurance tools to evaluate the risk of bias in the included studies. Since autism is highly heritable [77, 78], parent-mediated interventions are likely to contribute to parental outcomes alongside child-level outcomes [66, 79]. However, the research questions for the review focus on child-level outcomes only (social communication and language). This means that parental outcomes are not relevant to the RQs and so have not been included. Future systematic reviews could focus on research questions that address the effectiveness of parent-mediated play-based interventions on parental outcomes as these are valid and significant RQs.

## Supporting information

**S1 File. PRISMA-P checklist, search strings, data extraction form, quantified outcomes, and risk of bias assessment tool.**
(DOC)

## Author Contributions

**Conceptualization:** Gill Francis, Umar Toseeb.

**Investigation:** Emre Deniz, Gill Francis, Carole Torgerson, Umar Toseeb.

**Methodology:** Emre Deniz, Gill Francis, Carole Torgerson, Umar Toseeb.

**Supervision:** Carole Torgerson, Umar Toseeb.

**Validation:** Emre Deniz, Gill Francis, Carole Torgerson, Umar Toseeb.

**Writing – original draft:** Emre Deniz.

**Writing – review & editing:** Gill Francis, Carole Torgerson, Umar Toseeb.

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
