## [Decision Letter · Decision Letter 0]

10 Mar 2022

PONE-D-22-03826Parent-mediated play‐based interventions to improve social, language, and communication skills of preschool autistic children: A systematic review and meta-analysis protocolPLOS ONE

Dear Dr. Toseeb,

Thank you for submitting your manuscript to PLOS ONE. After careful consideration, we feel that it has merit but does not fully meet PLOS ONE’s publication criteria as it currently stands. Therefore, we invite you to submit a revised version of the manuscript that addresses the points raised during the review process.

I agree with the two reviewers that your paper has many strengths but may also benefit from a revision. Please consult the reviewers' comments which are straightforward and to which I have nothing to add. 

We look forward to receiving your revised manuscript.

Kind regards,

Robert Didden

Academic Editor

PLOS ONE

Journal Requirements:

Reviewers' comments:

Reviewer's Responses to Questions

**Comments to the Author**

1. Does the manuscript provide a valid rationale for the proposed study, with clearly identified and justified research questions?

Reviewer #1: Yes

Reviewer #2: No

2. Is the protocol technically sound and planned in a manner that will lead to a meaningful outcome and allow testing the stated hypotheses?

Reviewer #1: Yes

Reviewer #2: Yes

3. Is the methodology feasible and described in sufficient detail to allow the work to be replicable?

Reviewer #1: Yes

Reviewer #2: Yes

4. Have the authors described where all data underlying the findings will be made available when the study is complete?

Reviewer #1: No

Reviewer #2: Yes

5. Is the manuscript presented in an intelligible fashion and written in standard English?

Reviewer #1: Yes

Reviewer #2: Yes

6. Review Comments to the Author

You may also provide optional suggestions and comments to authors that they might find helpful in planning their study.

Reviewer #1: Thank you for the opportunity to review the study protocol entitled: “Parent-mediated play‐based interventions to improve social, language, and communication skills of preschool autistic children: A systematic review and meta-analysis.” The authors are correct that reviews have not yet focussed specifically on parent-implemented play-based intervention. Thus, the review that this protocol pertains to is likely to be important and informative.

This protocol has many strengths. It is well written and the rationale for the project is clear. The authors also intend to follow best practice guidelines in conducting the review, including independent agreement checks for the search, extraction, and quality indicators. The use of Cochrane RoB tools is also gold standard.

I do have several suggestions that could strengthen this manuscript.

Overarching feedback

1. I suggest that the authors use the term “autism” rather than ASD through the manuscript, except when referring specifically to the official diagnosis. This is because the autism community does not like the term “ASD”. This also aligns better with the use of identity first language elsewhere in the manuscript.

Introduction

2. Page 2- paragraph 1- It is interesting to me that play is defined as “ungoverned by rules” as many games that children play do have rules?

3. Page 2, paragraph 2- I am not sure if it necessary to mention Freud and psychoanalytic theories here since this is not an evidence-based approach.

4. Page 3, paragraph 2- A review of JASPER as recently been published, which the authors might consider including: Waddington, H., Reynolds, J. E., Macaskill, E., Curtis, S., Taylor, L. J., & Whitehouse, A. J. (2021). The effects of JASPER intervention for children with autism spectrum disorder: A systematic review. Autism, 25(8), 2370-2385.

5. It could be helpful to discuss the evidence for parent mediated intervention more generally, referencing recent reviews, such as the following: Nevill, R. E., Lecavalier, L., & Stratis, E. A. (2018). Meta-analysis of parent-mediated interventions for young children with autism spectrum disorder. Autism, 22(2), 84-98.

6. Page 3, final paragraph- The authors should make it clear that this is merely the protocol rather than the full research study.

7. Page 3, research questions- question one should specify that this relates to parent-mediated play-based interventions.

Methods

The methods were generally well-written and clear. I did have several areas of clarification in some sections, so have phrased these as questions.

1. Target sample- do all children in the study need to be under 6 years, or would a mean of 6 years be allowed? Will children showing high likelihood of autism, without an official diagnosis be included or excluded?

2. Intervention characteristics- How was Rubin et al.’s criteria be applied in order to determine whether an approach is considered “play-based”? Will the authors use a checklist? If so, will the authors get agreement on this? It would also be helpful to include this checklist as a supplementary document. Will routines-based interventions (e.g. ESDM), that use play often but not exclusively, be excluded? What constitutes a “play setting”?

3. Predictor and outcome variables- will the authors also examine parent outcome variables, for example their health and wellbeing, improvements in fidelity/ability to use the intervention techniques? This is important but often overlooked in reviews and meta-analyses.

4. Inclusion and exclusion criteria- do all children in the sample need to be diagnosed with ASD? That is, would you exclude studies that included some children “at high likelihood”? Would you include studies that had other implementers, such as a study delivered by both parents and teachers? If so, this could be a limitation, as it would not be possible to determine whether the effects were due to the parent or other implementer.

5. Study selection-phase 1: Have the authors considered doing a grey literature search? This is now considered best practice to reduce publication bias (Aromataris & Munn, 2020).

6. Data analysis and synthesis- Will any sensitivity analyses be conducted? For example. I wonder about separate analysis for the different control group types, depending on the number of relevant studies.

Discussion

1. It is not clear to me why the authors do not intend to focus also on parent outcomes? This does seem like something the authors could do, rather than stating it as a limitation of the study?

Grammatical, spelling, and referencing issues.

1. Although the manuscript is generally well written, I recommend that the authors do a thorough check for spelling and grammatical issues.

2. Page 3, lines 48-49: consider replacing “but” with “however” in the following sentence: “But there is considerable heterogeneity; affected individuals have varied profiles of strengths and difficulties.”

Reference

Aromataris, E., & Munn, Z. (2020). JBI manual for evidence synthesis. Retrieved from https://synthesismanual.jbi.global. https://doi.org/10.46658/JBIMES-20-01

Reviewer #2: This systematic review and meta-analysis looked at the key characteristics of play-based interventions and whether play-based interventions are more effective than non-play-based interventions. It also examined the factors influencing the effectiveness of play-based interventions on social, language and communication skills. Examples of play-based interventions are PRT, CCPT, and ASAP.

The study was overall well-planned and comprehension. The inclusion and exclusion criteria were clearly indicated. However, the authors should explain parent-mediated and its play-based interventions in the literature review. They should also justify why conducting this meta-analysis is important. The authors mentioned that no meta-analyses have been conducted in this regard. But why is it important? how is parent-mediated play-based intervention compared to non-parent-mediated play-based interventions? In order to address the importance or even key characteristics of parent-mediated play-based intervention, the authors should discuss how these characteristics are different from non-parent (e.g., therapists, siblings, teachers)

7. PLOS authors have the option to publish the peer review history of their article (what does this mean?). If published, this will include your full peer review and any attached files.

Reviewer #1: No

Reviewer #2: No

---

## [Author Response · Author response to Decision Letter 0]

22 Apr 2022

Journal: PLOS ONE

Submission-ID: PONE-D-22-03826

First Submission Date: 07/02/2022

Re-Submission date: 20/04/2022

Dear Dr Robert Didden,

Thank you for sending us the reviewers' comments on the submission of our review paper entitled ‘Parent-mediated, play‐based interventions to improve social communication and language skills of preschool autistic children: A systematic review and meta-analysis protocol”

We are thankful for your revise and resubmit decision on our manuscript. We found the reviewers’ comments useful and acknowledge that they further improved our manuscript. We have addressed the reviewers’ comments in our manuscript and/or in the response to the reviewer form which is summarised in the below table. In the table, we explained how we have addressed each comment and signposted the reviewers to the relevant section in the paper accordingly. In the re-submitted manuscript, we have also highlighted all the changes in yellow.

We hope the reviewers and you, as the academic editor, find our response satisfactory and are satisfied that our paper meets the standard for publication in your journal. 

Sincerely,

Dr Umar Toseeb

Table Summarising Response to Reviewers’ Comments

Reviewers’ comments Authors’ Responses

Reviewer 1 

Introduction 

1- Page 2- paragraph 1- It is interesting to me that play is defined as “ungoverned by rules” as many games that children play do have rules? We agree with the reviewer’s comment and acknowledge that the phrase was not accurately cited. It should have read “ungoverned by external rules”.

See page 2- paragraph 1.

2- Page 2, paragraph 2- I am not sure if it is necessary to mention Freud and psychoanalytic theories here since this is not an evidence-based approach. We welcome the reviewer’s comment. As suggested, we have replaced the mentioned paragraph from our manuscript with an additional paragraph on how play is associated with various developmental areas (cognitive, social communication, language, emotional, and physical) in young children. 

See page 2- paragraph 2.

3- Page 3, paragraph 2- A review of JASPER has recently been published, which the authors might consider including Waddington, H., Reynolds, J. E., Macaskill, E., Curtis, S., Taylor, L. J., & Whitehouse, A. J. (2021). The effects of JASPER intervention for children with autism spectrum disorder: A systematic review. Autism, 25(8), 2370-2385. We appreciate the reviewer’s comment and are thankful for drawing our attention to this paper which we were not previously aware of. As suggested, we have included the mentioned reference in the relevant paragraph alongside its additional findings. 

See page 3- paragraph 2 (Ref 39).

4- It could be helpful to discuss the evidence for parent-mediated intervention more generally, referencing recent reviews, such as the following: Nevill, R. E., Lecavalier, L., & Stratis, E. A. (2018). Meta-analysis of parent-mediated interventions for young children with autism spectrum disorder. Autism, 22(2),84-98. We appreciate the reviewer's suggestion. To address this comment, as well as the second reviewer’s first comment, we have included two new paragraphs into the introduction where we discussed the overall effectiveness of parent-mediated interventions and parent-mediated play-based interventions. We have also included this study in our “citation search review papers” to do backward chaining and identify any additional studies that meet our inclusion criteria. We are grateful to the reviewer for their suggestion of including a very relevant study in our manuscript. 

See page 4- paragraph 2 (Ref 55).

5. Page 3, final paragraph- The authors should make it clear that this is merely the protocol rather than the full research study. We agree with the reviewer. As suggested, we have made the following changes to the manuscript. 

- We have added the word “protocol” to our study title.

- We have revised the final paragraph of the introduction and highlighted that this is merely a protocol of a proposed systematic review and meta-analysis study.

See page 4- paragraph 2.

6. Page 3, research questions- question one should specify that this relates to parent-mediated play-based interventions. We agree with the reviewer. Research question 1 has been revised as follows: 

1. What are the key characteristics of parent-mediated play-based interventions?

See page 4- research question 1.

Methods 

7- Target sample- do all children in the study need to be under 6 years, or would a mean of 6 years be allowed? Will children showing the high likelihood of autism, without an official diagnosis be included or excluded?

 Age

- This is something we have considered prior to setting the age limit as an inclusion criterion. The existing literature suggests that the early years are crucial for social communication and language development in children’s lives. Additionally, researchers have also shown that children with impaired social communication and language skills in their early years are likely to continue to carry such impairments later in their life if not supported in the early years. We have also taken the school starting age into account when specifying this age limit as, in most countries, children usually start primary school (Year 1) at around the age of 7. Since our aim is to focus on parent-mediated play-based interventions in non-educational settings, we specified our target sample as preschool autistic children aged 6 years or younger. To address this comment, we revised our inclusion and exclusion criteria as follows:

Inclusion criteria

- Studies targeting preschool children aged six years old (72 months) or younger, 

Exclusion criteria

- Studies will be excluded if the upper age limit of their sample was greater than six years old (72 months) unless the authors separately reported the outcomes of those aged six years old or younger.

- 

See pages 7 & 8- inclusion and exclusion criteria.

Autism Diagnosis

The target sample of the current review is preschool autistic children. Therefore, an exclusion criterion was defined to make sure the included studies were conducted with autistic samples: Studies will be excluded if the participants in the sample were not formally diagnosed with or screened for autism ASD. As mentioned in the inclusion criteria, studies will be eligible in case their sample had a previous clinical diagnosis for autism or the researchers used valid autism screening measures to report the autistic characteristics of their sample. Studies will be excluded if their samples were not formally diagnosed with autism or screened for autism. To address this comment, the inclusion and exclusion criteria have been revised as follows:

Inclusion criteria

- Studies with a sample of autistic children or children with a previously diagnosed form of autism (e.g., Asperger’s syndrome, PDD-NOS, autistic spectrum disorder).

Exclusion criteria

- Studies will be excluded if the participants in the sample were not formally diagnosed with or screened for autism.

See pages 7 & 8- inclusion and exclusion criteria.

8- Intervention characteristics- How were Rubin et al.’s criteria applied in order to determine whether an approach is considered “play-based”? Will the authors use a checklist? If so, will the authors get an agreement on this? It would also be helpful to include this checklist as a supplementary document. Will routines-based interventions (e.g. ESDM), that use play often but not exclusively, be excluded? What constitutes a “play setting”? We appreciate the reviewer’s comment and agree that we should further clarify how Rubin et al.’s criteria will be applied to decide whether an intervention is play-based within the scope of our research. To address this, we have revised the intervention characteristics section as follows: 

Additionally, taking Rubin et al. 's (6) play approach into account, interventions that were not solely based on play, delivered during the playtime or within the playground will be excluded from this review. The latter criterion will be met in the following conditions: 

I. The intervention is a known and named play-based intervention (e.g, JASPER, PLAY, etc).

II. The intervention is a known and named therapeutic approach that uses play as a therapeutic tool (e.g., play therapy, child-centred play therapy, cognitive behavioural play therapy, Rogerian play, etc).

III. The intervention is a known and named developmental intervention that is built on the elements of play or consists of play-based activities such as joint attention, joint engagement, parent-child play (DIR Floortime, Pivotal Response Treatment, Early Start Denver Model, etc).

IV. The intervention is not a previously validated and named intervention but is judged as a play-based intervention by the authors based on the characteristics of the intervention. 

The last criterion will be met if “play is explicitly mentioned” (61) or if the intervention consists of parent-child play or child-led play activities. That is, any interventions that are not solely based on play are judged as not play-based (behavioural modification techniques, teaching and training activities, gaming, etc) will be excluded from the sample. Such interventions may include but are not limited to the following: Video modelling, video prompting, activity schedules, script fading, behavioural interventions, music therapy, virtual reality, gaming, and computer/video games, tablet applications.

See page 6- intervention characteristics.

In addition, we have also included the following exclusion criteria in our methods section to clearly state that the proposed review will not compare the effectiveness of two or more play-based activities. 

Exclusion criteria

- Studies that compared the effectiveness of two or more play-based interventions will be excluded unless they provide a control group that received a nonplay-based intervention.

- 

See page 8- exclusion criteria.

9- Predictor and outcome variables- will the authors also examine parent outcome variables, for example, their health and wellbeing, improvements in fidelity/ability to use the intervention techniques? This is important but often overlooked in reviews and meta-analyses.

 The authors agree with the reviewer in the sense that parental outcomes are often overlooked in systematic reviews of play literature. This is something we have considered, thus, we previously stated that parent-mediated play interventions are likely to contribute to parental outcomes alongside child-level outcomes in our protocol. Although we are aware that parent-mediated play-based interventions are likely to contribute to parental outcomes, alongside child’s outcomes, the main aim of our study is to report the effectiveness of parent-mediated play-based interventions on the social communication and language skills of preschool autistic children. Since this is a pre-registered protocol, we do not wish to go beyond the scope of our pre-defined research questions. However, we acknowledge that this is an important area and the implications for parental outcomes should be investigated and reported in future research. To address the reviewer’s comment, we have revised the relevant section in our limitations and suggested that future research could focus on parental outcomes as they are valid and significant. 

- Since autism is highly heritable (77, 78), parent-mediated interventions are likely to contribute to parental outcomes alongside child-level outcomes (79, 80). However, the research questions for the review focus on child-level outcomes only (social communication and language). This means that parental outcomes are not relevant to the RQs and so have not been included. Future systematic reviews could focus on research questions that address the effectiveness of parent-mediated play-based interventions on parental outcomes as these are valid and significant RQs.

See page 13- paragraph 2. 

Regarding outcome variables, we have also added a new section to our protocol indicating that autistic traits will be reported in our review alongside social communication and language outcomes.

- Additionally, autistic traits will also be included in the outcome variables if one or more studies reported autistic traits (e.g., restricted repetitive behaviours, social affect, calibrated severity score, etc) as primary outcome variables.

See page 7- predictor and outcome variables.

10- Inclusion and exclusion criteria- do all children in the sample need to be diagnosed with ASD? That is, would you exclude studies that included some children “at high likelihood”? Would you include studies that had other implementers, such as a study delivered by both parents and teachers? If so, this could be a limitation, as it would not be possible to determine whether the effects were due to the parent or other implementer.

 Autism Diagnosis

We appreciate the reviewer’s comment. This is something we have considered. The target sample of the current review is preschool autistic children. To address this comment, the inclusion and exclusion criteria have been revised as follows:

Inclusion criteria

- Studies with a sample of autistic children or children with a previously diagnosed form of autism (e.g., Asperger’s syndrome, PDD-NOS, autistic spectrum disorder).

Exclusion criteria

- Studies will be excluded if the participants in the sample were not formally diagnosed with or screened for autism.

As mentioned in the inclusion/exclusion criteria, studies will be eligible in case their sample had a previous clinical diagnosis for autism or the researchers used valid autism screening measures to report the autistic characteristics of their sample. Studies will be excluded if their sample did not have any clinical diagnosis of autism or screened for autism.

See page 7-inclusion & page 8-exclusion criteria.

Multiple implementers

As our study focuses on parent-mediated play-based interventions, studies will likely have professional experts (e.g., play therapists, speech and language therapists, clinicians, etc) who deliver the intervention to parents, and in turn, parents deliver the intervention to children. Since our study is exclusive to educational settings, any intervention that was conducted in educational settings will not meet our eligibility criteria and, therefore, be excluded. Therefore, we assume that there would be no teacher-mediated interventions in our sample. However, in any case, if there is more than one mediator in the intervention (parent/sibling/another caregiver), the study will be included only if the parent-mediated intervention is separately delivered alongside a non-play control group. To address this comment, we have revised our inclusion and exclusion criteria as follows:

Inclusion criteria

- Studies including at least one parent as a mediator. Studies with multiple implementers will also be included if implementers are placed in separate intervention groups alongside a non-play control group (e.g., intervention group 1= parent-mediated intervention, intervention group 2= sibling-mediated intervention, and a control group).

Exclusion criteria

- Studies will be excluded if there are multiple implementers in a single intervention group (e.g., intervention group= parent and sibling mediated play intervention and control group). 

See page 7-inclusion & page 8-exclusion criteria.

11- Study selection-phase 1: Have the authors considered doing a grey literature search? This is now considered best practice to reduce publication bias (Aromataris & Munn, 2020). Grey literature - Publication bias

We value the reviewer’s comment. We agree that doing a grey literature search is important for reducing publication bias. Although grey literature is often excluded from large databases, our search strategy was designed to cover grey literature. As can be seen in our PRISMA flow diagram, we searched ERIC, ProQuest, PsycINFO, and Google scholar database which provide extensive grey literature including but not limited to “pre-registered controlled trials, doctoral (PhD) dissertations, conference papers, preprints (not yet peer-reviewed and/or published) studies. Our initial stage literature search (n=100) indicated that some of the grey literature made it into our sample (PhD dissertations). Therefore, we are confident that our extensive literature search did not suffer from publication bias. To make this clearer to the reader, we have reworded the following inclusion criteria: 

Inclusion criteria

- Studies that are published between 2000 and 2021 including grey literature. 

See page 7-inclusion & exclusion criteria.

12- Data analysis and synthesis- Will any sensitivity analyses be conducted? For example. I wonder about separate analysis for the different control group types, depending on the number of relevant studies. We agree with the reviewer’s comment. To address this comment, we have included an additional statement in our analysis section: 

- Furthermore, sensitivity analysis may be conducted to assess whether the methodological quality of studies (e.g., low risk of bias, some concerns) showed any impact on the reported effect size of the interventions. Additional subgroup sensitivity analyses may also be used to check if intervention effects vary by research design (RCT or QED) if there is a sufficient number of relevant studies.

See page 11- paragraph 3.

Discussion 

13- It is not clear to me why the authors do not intend to focus also on parent outcomes? This does seem like something the authors could do, rather than stating it as a limitation of the study?

 This is something we have considered. In our protocol, we previously stated that parent-mediated play interventions are likely to contribute to parental outcomes alongside child-level outcomes. However, we aim to focus only on child-level outcomes in the proposed review. To address the reviewer’s comment, we have revised the relevant section in our limitations and suggested that future research could focus on parental outcomes as they are valid and significant.

- Since autism is highly heritable (42), parent-mediated interventions are likely to contribute to parental outcomes alongside child-level outcomes (43). However, the research questions for the review focus on child-level outcomes only (social communication and language). This means that parental outcomes are not relevant to the RQs and so have not been included. Future SRs could focus on research questions that address the effectiveness of parent-mediated play-based interventions on parental outcomes as these are valid and significant RQs.

See Page 13, paragraph 2.

Overarching feedback 

14- I suggest that the authors use the term “autism” rather than ASD through the manuscript, except when referring specifically to the official diagnosis. This is because the autism community does not like the term “ASD”. This also aligns better with the use of identity-first language elsewhere in the manuscript. The authors of this manuscript follow the neurodiversity movement. In this sense, we agree with the reviewer that the term “ASD” does not align well with the neurodiversity approach as well as the autistic community’s views. Therefore, as suggested, we have revised the manuscript and replaced the term “ASD” with “autism” throughout the manuscript. We thank the reviewer’s contribution.

15- Although the manuscript is generally well written, I recommend that the authors do a thorough check for spelling and grammatical issues. Thanks for the reviewer’s attention to the spelling and grammatical errors. We proofread the manuscript thoroughly and corrected all spelling and grammatical errors. 

16- Page 3, lines 48-49: consider replacing “but” with “however” in the following sentence: “But there is considerable heterogeneity; affected individuals have varied profiles of strengths and difficulties.” As suggested, we have replaced the word “but” with “however” in the mentioned sentence. 

Please see Page 3 paragraph 1. 

Reviewer 2 

Comment Authors’ Responses

1- The study was overall well-planned and comprehension. The inclusion and exclusion criteria were clearly indicated. However, the authors should explain parent-mediated and its play-based interventions in the literature review. We appreciate the reviewer’s comment. To address this, we have included two new paragraphs explaining the characteristics and effectiveness of parent-mediated play-based interventions in the introduction.

- Regarding the effectiveness of parent-mediated play-based interventions, JASPER has been found effective in increasing the joint engagement and play skills of preschool autistic children (46). Additionally, the Play and Language for Autistic Youngsters (PLAY) intervention significantly improved social and emotional functioning and reduced autism severity in preschool autistic children (57). In another study, a one-year Focused Playtime Intervention (FPI) has been reported to have significant positive effects on the expressive language skills of preschool autistic children (58). A pilot study has reported that a parent-mediated Developmental, Individual-differences, Relationship-based model (DIR/Floortime) intervention significantly improved the functional and emotional skills and core autistic traits of preschool autistic children (59). Although individual studies reported the effectiveness of parent-mediated play-based interventions, to the best of the authors’ knowledge, no previous systematic review or meta-analysis has reported the characteristics and effectiveness of parent-mediated play-based interventions in preschool autistic children. Thus, the characteristics and overall effectiveness of such interventions remain unknown.

See page 4- paragraph 1. 

2- They should also justify why conducting this meta-analysis is important. The authors mentioned that no meta-analyses have been conducted in this regard. But why is it important? how is parent-mediated play-based intervention compared to non-parent-mediated play-based interventions? In order to address the importance or even key characteristics of parent-mediated play-based intervention, the authors should discuss how these characteristics are different from non-parent (e.g., therapists, siblings, teachers) We agree with the reviewer in the sense that there was a need to better explain why this study is needed in the existing literature. To address this, we have included two new paragraphs into our introduction and explained the characteristics and aim of peer-mediated, teacher-mediated, and parent-mediated interventions in addition to the paragraph on parent-mediated-play-based interventions (Page 5 Paragraph 1. )

- While the majority of play-based interventions have solely been built on the interaction between a child and interventionist, there has been a recent trend in the inclusion of mediators such as peers, teachers, and parents. This, perhaps, is not only due to the mediators' ability and availability in mediating such interventions but also targeting an improvement in both children’s and mediators’ outcomes within one intervention (46). Unlike parent-mediated interventions, peer- and teacher-mediated interventions are often based on group play activities, conducted in educational settings, and are built on the elements of role-play and social reinforcement (47, 48). Both peer and teacher-mediated interventions have been reported to be significant in improving social communication and language skills (49-51), as well as core autistic traits (52).

- Parent-mediated interventions, however, differ from peer- and teacher-mediated interventions in the sense that they are often one-to-one activities between a parent and a child, take place in non-educational settings (e.g., home, clinics), and are commonly based on the principles of Mediated Learning Theory (53). Parent-mediated interventions have been found to have significant positive effects on certain developmental skills of autistic children such as joint attention, social interaction, language, nonverbal communication, and adaptive functioning skills (54). Additionally, a recent meta-analysis has reported a significant effect size for parent-mediated interventions on the communication and language skills of preschool autistic children (55). Similarly, some researchers have found that parent-mediated interventions are effective in improving language comprehension and child-parent interaction, as well as decreasing autism severity in young autistic children (56). 

See page 3-paragraph 2 & 3.

---

## [Decision Letter · Decision Letter 1]

26 Apr 2022

PONE-D-22-03826R1Parent-mediated play‐based interventions to improve social communication and language skills of preschool autistic children: A systematic review and meta-analysis protocolPLOS ONE

Dear Dr. Toseeb, I have received the reviews on your first revised manuscript from the two reviewers and I have read the revision myself. It has much improved and the paper is almost ready for publication. Reviewer #1 has one last comment that should be addressed before I can accept the paper. It should not be a problem to revise the paper a second time.

We look forward to receiving your revised manuscript.

Kind regards,

Robert Didden

Academic Editor

PLOS ONE

Journal Requirements:

Reviewers' comments:

Reviewer's Responses to Questions

**Comments to the Author**

1. Does the manuscript provide a valid rationale for the proposed study, with clearly identified and justified research questions?

Reviewer #1: Yes

Reviewer #2: Yes

2. Is the protocol technically sound and planned in a manner that will lead to a meaningful outcome and allow testing the stated hypotheses?

Reviewer #1: Yes

Reviewer #2: Yes

3. Is the methodology feasible and described in sufficient detail to allow the work to be replicable?

Reviewer #1: Yes

Reviewer #2: Yes

4. Have the authors described where all data underlying the findings will be made available when the study is complete?

Reviewer #1: No

Reviewer #2: Yes

5. Is the manuscript presented in an intelligible fashion and written in standard English?

Reviewer #1: Yes

Reviewer #2: Yes

6. Review Comments to the Author

You may also provide optional suggestions and comments to authors that they might find helpful in planning their study.

Reviewer #1: I would like to thank the authors for their careful attention to the edits. I am very satisfied with these edits and only have one small question remaining, which pertains to the inclusion/exclusion. The authors have stated that children who were not screened for autism will be excluded but there is no mention of positive screening in the inclusion criteria. Therefore, it is not clear to me whether children screened as "high likelihood of autism" but not yet diagnosed will be included. If they are, the inclusion sentence could read as follows: "Studies with a sample of autistic children, children who screened as "high likelihood for autism" on a validated screening too, or children with a previously diagnosed form of autism (e.g., Asperger’s syndrome, PDD-NOS, autistic spectrum disorder). If they will be excluded, then the mention of screening in the exclusion criteria can be removed.

Reviewer #2: I appreciate the authors had provided more explanations and justifications on the rationale of parent-mediated intervention and why it is important.

7. PLOS authors have the option to publish the peer review history of their article (what does this mean?). If published, this will include your full peer review and any attached files.

Reviewer #1: No

Reviewer #2: No

---

## [Author Response · Author response to Decision Letter 1]

28 Apr 2022

Reviewer #1: I would like to thank the authors for their careful attention to the edits. I am very satisfied with these edits and only have one small question remaining, which pertains to the inclusion/exclusion. The authors have stated that children who were not screened for autism will be excluded but there is no mention of positive screening in the inclusion criteria. Therefore, it is not clear to me whether children screened as "high likelihood of autism" but not yet diagnosed will be included. If they are, the inclusion sentence could read as follows: "Studies with a sample of autistic children, children who screened as "high likelihood for autism" on a validated screening too, or children with a previously diagnosed form of autism (e.g., Asperger’s syndrome, PDD-NOS, autistic spectrum disorder). If they will be excluded, then the mention of screening in the exclusion criteria can be removed.

RESPONSE:We appreciate the reviewer’s positive comments and are thankful for their very valuable contribution to our manuscript.

Autism Diagnosis

We agree with the reviewer’s point in the sense that the mentioned inclusion criteria should have been further clarified. To address this comment, we revised our inclusion criteria as follows:

Inclusion criteria

- Studies with a sample of autistic children, including the ones who screened as "high likelihood for autism" on a validated autism screening measure, or children with a previously diagnosed form of autism (e.g., Asperger’s syndrome, PDD-NOS, autistic spectrum disorder). See page 10, inclusion criteria 2

Reviewer #2: I appreciate the authors had provided more explanations and justifications on the rationale of parent-mediated intervention and why it is important.

RESPONSE:We appreciate the reviewer’s positive comments and are thankful for their contribution to our manuscript. 

Additional Revisions: The reference list has been checked for typing errors and all typing errors have been corrected. In addition, the following reference has been corrected: 6. Rubin KH, Fein GG, Vandenberg B. Play In PH Mussen & EM Hetherington (Eds.), Handbook of child psychology vol. 4: socialization, personality, and social development (Vol. 4, pp. 693-774). New York; Chichester: Wiley, 1983.

---

## [Decision Letter · Decision Letter 2]

6 Jun 2022

Parent-mediated play‐based interventions to improve social communication and language skills of preschool autistic children: A systematic review and meta-analysis protocol

PONE-D-22-03826R2

Dear Dr. Toseeb,

We’re pleased to inform you that your manuscript has been judged scientifically suitable for publication and will be formally accepted for publication once it meets all outstanding technical requirements.

Kind regards,

Robert Didden

Academic Editor

PLOS ONE

Additional Editor Comments (optional):

Reviewers' comments:

Reviewer's Responses to Questions

**Comments to the Author**

1. Does the manuscript provide a valid rationale for the proposed study, with clearly identified and justified research questions?

Reviewer #1: Yes

2. Is the protocol technically sound and planned in a manner that will lead to a meaningful outcome and allow testing the stated hypotheses?

Reviewer #1: Yes

3. Is the methodology feasible and described in sufficient detail to allow the work to be replicable?

Reviewer #1: Yes

4. Have the authors described where all data underlying the findings will be made available when the study is complete?

Reviewer #1: No

5. Is the manuscript presented in an intelligible fashion and written in standard English?

Reviewer #1: Yes

6. Review Comments to the Author

You may also provide optional suggestions and comments to authors that they might find helpful in planning their study.

Reviewer #1: Thank you very much for your careful attention to my suggested revision. The inclusion criteria are now very clear.

7. PLOS authors have the option to publish the peer review history of their article (what does this mean?). If published, this will include your full peer review and any attached files.

Reviewer #1: No

---

## [Editor Report · Acceptance letter]

5 Aug 2022

PONE-D-22-03826R2 

Parent-mediated play‐based interventions to improve social communication and language skills of preschool autistic children: A systematic review and meta-analysis protocol 

Dear Dr. Toseeb:

I'm pleased to inform you that your manuscript has been deemed suitable for publication in PLOS ONE. Congratulations! Your manuscript is now with our production department. 

Kind regards, 

on behalf of

Professor Robert Didden 

Academic Editor

PLOS ONE